# Neuropathological Applications of Microscopy with Ultraviolet Surface Excitation (MUSE): A Concordance Study of Human Primary and Metastatic Brain Tumors

**DOI:** 10.3390/brainsci14010108

**Published:** 2024-01-22

**Authors:** Mirna Lechpammer, Austin Todd, Vivian Tang, Taryn Morningstar, Alexander Borowsky, Kiarash Shahlaie, John A. Kintner, John D. McPherson, John W. Bishop, Farzad Fereidouni, Zachary T. Harmany, Nicholas Coley, David Zagzag, Jason W. H. Wong, Jiang Tao, Luke B. Hesson, Leslie Burnett, Richard Levenson

**Affiliations:** 1Department of Pathology and Laboratory Medicine, University of California Davis Health, Sacramento, CA 95817, USA; austinwtodd@gmail.com (A.T.); vivian.tang@ucsf.edu (V.T.); morningt@uci.edu (T.M.); adborowsky@ucdavis.edu (A.B.); jakintner@ucdavis.edu (J.A.K.); jwabishop@ucdavis.edu (J.W.B.); fereidouni@ucdavis.edu (F.F.); zharmany@gmail.com (Z.T.H.); nbcoley@gmail.com (N.C.); rmlevenson@ucdavis.edu (R.L.); 2Department of Biochemistry and Molecular Pharmacology, New York University Langone Medical Center, New York, NY 10016, USA; 3Pathology and Laboratory Operations, Foundation Medicine, Inc., Cambridge, MA 02141, USA; 4Department of Neurosurgery, University of California Davis Health, Sacramento, CA 95817, USA; krshahlaie@ucdavis.edu; 5Department of Biochemistry and Molecular Medicine, University of California Davis Health, Sacramento, CA 95817, USA; jdmcpherson@ucdavis.edu; 6Departments of Pathology and Neurosurgery, New York University Langone Medical Center, New York, NY 10016, USA; david.zagzag@nyulangone.org; 7School of Biomedical Sciences, Li Ka Shing Faculty of Medicine, The University of Hong Kong, Pok Fu Lam, Hong Kong SAR, China; jwhwong@hku.hk; 8Kinghorn Centre for Clinical Genomics, Garvan Institute of Medical Research, Darlinghurst NSW 2010, Australia; jiang.tao@bgi.com (J.T.); l.hesson@unsw.edu.au (L.B.H.); l.burnett@garvan.org.au (L.B.); 9School of Clinical Medicine, University of New South Wales Sydney, St Vincent’s Healthcare Clinical Campus, Darlinghurst NSW 2010, Australia; 10Department of Molecular Genetics, Douglass Hanly Moir Pathology, Macquarie Park NSW 2113, Australia; 11School of Clinical Medicine, University of New South Wales Sydney, Randwick NSW 2052, Australia

**Keywords:** microscopy, imaging, MUSE, brain neoplasm, neuropathology

## Abstract

Whereas traditional histology and light microscopy require multiple steps of formalin fixation, paraffin embedding, and sectioning to generate images for pathologic diagnosis, Microscopy using Ultraviolet Surface Excitation (MUSE) operates through UV excitation on the cut surface of tissue, generating images of high resolution without the need to fix or section tissue and allowing for potential use for downstream molecular tests. Here, we present the first study of the use and suitability of MUSE microscopy for neuropathological samples. MUSE images were generated from surgical biopsy samples of primary and metastatic brain tumor biopsy samples (n = 27), and blinded assessments of diagnoses, tumor grades, and cellular features were compared to corresponding hematoxylin and eosin (H&E) images. A set of MUSE-treated samples subsequently underwent exome and targeted sequencing, and quality metrics were compared to those from fresh frozen specimens. Diagnostic accuracy was relatively high, and DNA and RNA integrity appeared to be preserved for this cohort. This suggests that MUSE may be a reliable method of generating high-quality diagnostic-grade histologic images for neuropathology on a rapid and sample-sparing basis and for subsequent molecular analysis of DNA and RNA.

## 1. Introduction

The diagnosis of brain tumor biopsies has traditionally relied upon histology methods in which tissue must be processed, fixed, and sectioned to be ready for light microscopy imaging. Generally, a hematoxylin and eosin (H&E) slide requires overnight processing and, as such, involves costly delays in diagnosis. Specimens acquired using a core needle biopsy, furthermore, are small and will be diminished in volume due to these histopathological preparations, leaving less material available for further tests. This is becoming more of a dilemma as novel molecular characterization techniques such as next-generation sequencing and genome-wide methylation profiling emerge as components of precision medicine [1], and tissue allocation for these molecular tests is increasingly prioritized. Although brain biopsy procedures are generally considered safe, there are nevertheless significant risks involved in obtaining more tissue, including perioperative seizure, hemorrhage, infection [2], and death [3]. Thus, time, expense, and tissue consumption indicate a compelling need to address current intraoperative protocols with a validated and reliable sample-sparing, histology-scale imaging method.

Here we propose a fluorescence-based, slide-free optical imaging system, Microscopy Using Ultraviolet Excitation (MUSE), as a potential solution. One of the hallmarks of MUSE is the elimination of fixation and sectioning steps; a sample needs only be briefly stained and then gently flattened against an imaging window to allow the creation of a high-resolution mosaic image. The UV excitation of MUSE penetrates to a shallow depth of approximately 10 µm [4], allowing for a rapid analysis of the surface of unsectioned tissues (of any thickness) prior to subsequent embedding or sectioning, if desired. As demonstrated in prior studies with breast, pancreas, lung, thyroid, kidney [4], esophageal [5] (including Barrett’s esophagus) [6], and dermatopathological neoplasms [7], MUSE reliably generates diagnostic-grade images with exquisite resolution and enhanced visualization of surface topography. Here, we propose and pilot a study supporting the utility and applicability of MUSE for neuropathological diagnostics by: (a) comparing diagnostic accuracy with traditional H&E methods; and (b) assessing the DNA and RNA integrity of specimens following MUSE treatment. 

## 2. Materials and Methods

### 2.1. Image Acquisition and Diagnostic Accuracy Comparison

Tissue from cases of surgical resections of brain tumors (n = 27; mean age 55.5 years, with range 31–86 years; M:F ratio of 13:11) was selected to span a range of tumor types and anatomical structures; these included diffuse astrocytomas (n = 10), oligodendroglial tumors (n = 4), meningiomas (n = 6), metastatic tumors (n = 4), and normal anatomical structures (n = 3). A total of 67 images were generated from these cases. The tissues for this pilot study had been previously snap-frozen in liquid nitrogen, although MUSE can also be performed on fresh or formalin-fixed tissue. Once thawed, the tissue was bisected—one half was placed in phosphate-buffered saline (PBS) and then 10% formalin for eventual H&E staining, while the other half was stained with fluorescent dyes and imaged with MUSE. If necessary, the surfaces of the MUSE samples were flattened by careful cutting (by hand) with a scalpel and then stained as previously described [4] using a solution containing both Hoechst and rhodamine for approximately 10–30 s. A suitable combination proved to be rhodamine B (Sigma-Aldrich, Burlington, MA, USA; 500 μg/mL in PBS) plus Hoechst 33,342 (Life Technologies, Carlsbad, CA, USA; 500 μg/mL in PBS) combined in a single solution. A brief PBS wash followed. The surface to be imaged was then gently compressed against a sapphire window (typical glass does not transmit UV light in the range required for MUSE). Excitation of the fluorescently stained surface components was accomplished via oblique excitation using ultraviolet-emitting light-emitting diodes (LEDs) centered at a wavelength of 275–285 nm. The resulting broadband-emitted signals were then collected with a 10X, 0.45 NA lens and transmitted to a 9-megapixel RGB (color) CCD camera. Exposure times were typically in the range of 200–300 ms, resulting in a total scan time of less than 5 min per square-cm of tissue. After each 10X field (approximately 1 × 1 mm) was captured, the stage would shift to the next position; the captured fields were then mosaicked to create a composite image of the entire scanned area. Across all images, brightness and color density settings were preserved, as well as appropriate white-balancing and H&E color conversion through an open-source image editing program, GNU Image Manipulation Program (GIMP), as described [4]. At the conclusion of imaging, samples were snap-frozen to be processed for DNA and RNA extraction. One sample of cerebellar metastatic Mullerian adenocarcinoma was divided, with one half undergoing immunohistochemistry (IHC) staining with cytokeratins 2 and 20 and PAX-8 after UV light exposure to simulate MUSE imaging conditions, and the other half undergoing an identical IHC protocol without UV light.

H&E images were generated from the other half of the tissue samples that underwent traditional formalin fixation, paraffin embedding, and imaging by slide scanner (Aperio AT2, Leica Biosystems Imaging, Inc., Vista, CA, USA) on sites corresponding to those on the MUSE-generated images. A total of 67 paired images were generated.

The blinded concordance study involved a panel of three board-certified pathologists, including two board-certified neuropathologists (DZ, ML), and one board-certified surgical pathologist (JB). MUSE images were viewed digitally in random sequence, and diagnoses were written in clinical pathology report format, and a score of “correct” or “incorrect” was assigned based on equivalence or discrepancy compared to an established diagnosis made on the H&E image. In addition to rendering diagnoses, estimates of tumor grade, cellularity, nuclear atypia, and mitoses on each specimen were recorded for each MUSE tumor image. Table 1 displays the composite scores for correctly assigning a diagnostic category for the MUSE image (diffuse astrocytomas, oligodendroglial tumors, meningiomas, metastatic tumors, and normal anatomic structures), based on the number of cases examined. Table 2 displays the accuracy of grade assignments per image analyzed. Table 3 displays the accuracy of cellular features of tumors (where cellularity was designated as “mild/moderate” versus “dense”, atypia as “mild/moderate” versus “severe”, and mitoses as “present” versus “absent”), per image analyzed. The combined accuracy for each table was taken as a quotient of the sum of correct responses divided by the sum of total responses.

### 2.2. Whole-Exome Sequencing

DNA libraries were prepared from 200 ng genomic DNA samples using the KAPA HyperPrep kit (Roche, Basel, Switzerland, Cat. KK8504, 07962363001). Exome captures were performed using the Agilent SureSelect Clinical Research Exome V2 kit (CREv2, Agilent Technologies, Santa Clara, CA, USA, Cat. 5190-9493). qPCR-based quantification of captured libraries was performed with the KAPA Library Quantification Kit for Illumina Platform (Roche, Basel, Switzerland, Cat. KK4824, 07960140001) using a QuantStudio 7 real-time thermocycler (Thermo Fisher Scientific, Waltham, MA, USA). Indexed exome libraries were pooled and loaded at 150 pM into Illumina NovaSeq S1 and S4 flowcells using XP mode. Sequencing runs (2 × 150 cycles, dual indexes) were completed on an Illumina NovaSeq 6000 sequencer (Illumina, San Diego, CA, USA). Fastq files of raw sequencing reads were combined across runs for each sample for further analysis. Raw reads were aligned using BWA (0.7.17) [8] against the human reference genome, GRCh37. Duplicate reads were removed using Picard Tools (2.20) [http://broadinstitute.github.io/picard/] (accessed on 1 September 2019), and quality scores were recalibrated using GATK (3.8) [9]. For variant calling, two algorithms, Strelka2 (2.9.7) [10] and Pisces (5.2.11) [11], were both used with default parameters. A variant was considered genuine if it was called by both algorithms and had a variant allele frequency of >0.3. Only variants that fell within 10 bp of the target region were retained. Ensembl VEP (92.6) [12] was used for variant annotation.

### 2.3. Targeted Sequencing of DNA and RNA Variant Calling

Oncomine Comprehensive Assay v3 (OCAv3) is an amplicon-based panel targeting the DNA or RNA of 161 genes relevant to solid tumors. Libraries were prepared from 20 ng of DNA and 40 ng of RNA using the Ion AmpliSeq Library Kit Plus and the Oncomine Comprehensive Panel v3M (Thermo Fisher Scientific, Waltham, MA, USA, Oncomine™ Comprehensive Assay v3M, Cat. A36111). Each library was uniquely labelled with the IonCode Adapter (Thermo Fisher Scientific, Waltham, MA, USA, Cat. A29751). Libraries were quantified using the Ion Library TaqMan Quantitation Kit (Thermo Fisher Scientific, Waltham, MA, USA, Cat. 4468802) on a QuantStudio 7 real-time thermocycler (Thermo Fisher Scientific, Waltham, MA, USA), and 50 pmoles of each library were then pooled and sequenced by loading onto a 540 IonChip using an IonChef instrument (Thermo Fisher Scientific, Waltham, MA, USA). Ion semiconductor sequencing was completed using an S5 GeneStudio (400 flows, or equivalent to ~200 bp). Coding region missense and nonsense variants with a variant allele frequency >0.05 with ≥20 variant reads and total coverage of 400 reads were identified using Torrent Suite (v5.10.1) and Ion Reporter (v5.10.5.0) software. RNA data were assessed using intron-spanning exonic reads in control amplicons present in OCAv3. Read counts were normalized according to read counts per million reads.

## 3. Results

### 3.1. MUSE Image Quality Compared to Conventional Histology

As a whole, MUSE allowed for greatly expedited processing, as the average time required to generate one specimen image was approximately 5 min, compared to the overnight processing typically needed to generate an H&E-stain slide. Figure 1 illustrates how MUSE can preserve many architectural aspects of normal anatomical structures, as seen with the cerebellum, pineal gland, and Pacinian corpuscle. Figure 2, Figure 3 and Figure 4 apply the same principle to primary and metastatic tumors, where cellular features, large calcifications, and other histopathological aspects can be appreciated. Finally, Figure 5 suggests maintenance of IHC integrity on tissue that had previously undergone MUSE imaging, with no discernible difference in staining quality.

### 3.2. Preliminary Validation Studies

Overall, concordance between MUSE diagnoses and H&E diagnoses appeared robust, with a trend favoring greater concordance rates among more senior and specialized neuropathologists. Table 1 displays the accuracy of diagnoses rendered from MUSE images, with each MUSE diagnosis being scored as “correct” if it matched the tumor type assigned based on the H&E image. The data suggests a high degree of diagnostic accuracy, with an accuracy of 89% or higher within each tumor type. Each reviewer also individually demonstrated an overall accuracy of at least 89%, and combined, the reviewers achieved an overall accuracy of 94%. Table 2 displays the accuracy of grading CNS tumors from MUSE images. There appears to be an increased error rate within the category of high-grade tumors and increased inter-reviewer variability when comparing the total combined accuracy for each reviewer across tumor grades. Neuropathology can be diagnostically challenging for general anatomical pathologists, which is probably why there were more discordant cases for glioblastoma multiforme, anaplastic oligodendrogliomas, and grade 2 meningiomas. However, the accuracy for low-grade tumors remained promising, with 100% accuracy for diffuse astrocytomas and grade I meningiomas and 96% for oligodendrogliomas. Taken together, the overall accuracy for grading MUSE images was 73%. Table 3 displays the accuracy in determining cellularity, atypia, or mitoses in MUSE images of tumors by binary scale. Among the errors made, the most common trend observed was that cellularity and atypia were overestimated and mitoses underestimated on MUSE compared to H&E. Nonetheless, overall accuracy, as well as combined accuracy for all reviewers, was above 80% in this preliminary study, suggesting a high degree of reliability in MUSE images accurately depicting pathologic features.

**Table 1 brainsci-14-00108-t001:** Accuracy of tumor type diagnosis.

	Reviewer 1		Reviewer 2		Reviewer 3		
Tumor Type	Correct	Incorrect	Correct	Incorrect	Correct	Incorrect	Combined Accuracy
Diffuse Astrocytic DA II, AA III, and GBM IV	10	0	9	1	9	1	93%
Oligodendroglial Oligo II, AO III	4	0	4	0	3	1	92%
Meningioma Grades I and II	6	0	5	1	5	1	89%
Metastatic	3	0	3	0	3	0	100%
Normal Anatomic Structures	4	0	4	0	4	0	100%
**Combined** **Accuracy**	100%		93%		89%		94%

Note: Each score of “1” represents one specimen (ranging from 1 to 4 images for each). “Correct” is defined as all tumor type designations made on MUSE images matching those of the H&E. “Incorrect” is defined as one or more tumor type designation discrepancies. Combined accuracy was calculated as the quotient of the sum of correct responses over the sum of total responses.

**Table 2 brainsci-14-00108-t002:** Accuracy of tumor grading.

	Reviewer 1		Reviewer 2		Reviewer 3		
Tumor Type	Correct	Incorrect	Correct	Incorrect	Correct	Incorrect	Combined Accuracy
Diffuse Astrocytoma Grade II	5	0	5	0	5	0	100%
Anaplastic Astrocytoma Grade III	6	0	5	1	3	3	78%
Glioblastoma Multiforme Grade IV	10	6	9	7	10	6	60%
Oligodendroglioma Grade II	8	0	8	0	7	1	96%
Anaplastic Oligodendroglioma Grade III	2	0	0	2	0	2	33%
Meningioma Grade I	5	0	5	0	5	0	100%
Meningioma Grade II	4	1	0	5	1	4	33%
**Combined** **Accuracy**	85%		68%		66%		73%

Note: Each score of “1” represents one image. “Correct” is defined as the grading designation made on MUSE images matching that of the H&E, and “incorrect” is defined as the grading designation not matching. Combined accuracy was calculated as the quotient of the sum of correct responses over the sum of total responses.

**Table 3 brainsci-14-00108-t003:** Accuracy of assessment for cellular features.

	Reviewer 1		Reviewer 2		Reviewer 3		
Cellular Feature	Correct	Incorrect	Correct	Incorrect	Correct	Incorrect	Combined Accuracy
Cellularity	36	11	39	8	41	6	82%
Atypia	44	3	43	4	43	4	92%
Mitoses	38	9	34	13	33	14	74%
**Combined** **Accuracy**	84%		82%		83%		83%

Note: Each score of “1” represents one image. “Correct” is defined as the cellular feature description made on MUSE images matching that of the H&E, and “incorrect” is defined as the cellular feature description not matching. Combined accuracy was calculated as the quotient of the sum of correct responses over the sum of total responses.

### 3.3. Suitability for Downstream Molecular Analysis

A subset of MUSE-treated samples underwent additional pilot testing for DNA and RNA integrity following imaging. Table 4 displayed similar DNA and RNA yield purity from MUSE-treated (n = 2) versus fresh frozen tissues (n = 2). From there, two commonly used methods for preparing DNA for next-generation sequencing methods are amplicon-based and enrichment-based library preparation methods. Sequencing itself can also be performed using different sequencing chemistry, including fluorescence-based or ion semiconductor sequencing-by-synthesis. Table 5 and Table 6 compare quality-control metrics obtained following sequencing and alignment, showing that the volume and quality of sequencing data obtained from both types of tissue were comparable in regard to numbers of DNA and RNA sequencing reads, read length, and target coverage. Additionally, Table 5 includes an assessment of DNA damage in the form of the proportion of variants that were C>T transitions, as sequencing artifacts caused by formalin degradation are represented by an artifactual increase in C:G>T:A transitions in DNA [13]. Similar frequencies of C>T transitions were detected in both MUSE-treated and fresh frozen tissues, indicating that MUSE treatment does not generate the artifacts typically associated with formalin fixation. Artifacts from formalin fixation can make the bioinformatics pipelines required for molecular diagnostics more difficult and “noisy” compared to those obtained from fresh tissue and blood samples.

The amplicon-based assay we utilized also targets RNA. To assess whether MUSE treatment adversely affects the sequencing of RNA, we compared normalized read counts encompassing spliced exons across transcripts from seven genes. Abundant RNA read counts were detected in all transcripts (Appendix A). Furthermore, no major differences in read counts were detected in data derived from MUSE-treated versus fresh frozen tissues. Small differences were seen for some transcripts; however, these are likely related to differences in gene expression. Next, we assessed whether MUSE treatment affects the accuracy of sequencing. To do this, we compared sequence variants identified in MUSE-treated and paired fresh-frozen tissues. We focused our analysis on high-confidence variants located within exons and splice sites that altered amino acid sequences (see Section 2, Materials and Methods). Our analysis did not distinguish variants based on pathogenicity or whether they were germline or somatic. We identified the same variants in MUSE-treated versus paired fresh-frozen tissues using either amplicon or enrichment-based library preparation methods (Appendix A). 

## 4. Discussion

This study suggests that MUSE imaging confers a high degree of diagnostic accuracy in neuropathological tumor specimens while offering a more rapid and sample-sparing means of image acquisition. Combined accuracy rates of more than 80% were achieved for correct diagnosis of tumor type and characterizations of cellular features on MUSE images, with as high as 100% concordance observed for metastatic tumors and normal anatomical structures. The accuracy of grading on MUSE images was high for low-grade tumors (a range of 96–100%), albeit lower for high-grade tumors (33–78%). Additionally, preliminary data shows that MUSE treatment is compatible with amplicon and enrichment-based sequencing of both DNA and RNA and with next-generation sequencing methodologies commonly used in research and diagnostic laboratories. Given a 5-min protocol versus overnight processing and the preserved integrity of tissue as intact and unfixed, MUSE appears in many aspects advantageous compared to traditional microscopy methods.

This study is the first to analyze the applicability of MUSE microscopy with neuropathological specimens and to evaluate the ability of MUSE-treated samples to undergo downstream molecular diagnostics and sequencing. Previous work by Qorbani et al. [7] also analyzed concordance with H&E images for dermatologic specimens, although their study analyzed normal dermal structures and utilized a 0–2 scoring system (where 0 indicated MUSE not being able to identify structures without H&E correlation and 2 indicating MUSE being able to identify structures with greater certainty than H&E), compared to percent correct in our data analysis. Their total mean score of 9.4/10 agrees well with our 94% overall combined accuracy rate for correct diagnosis (Table 1). We are unaware of work published to date on molecular testing of MUSE-treated samples.

This study is also the first to analyze the ability to grade and identify cellular features with MUSE. We were surprised to find that MUSE images were perceived to have higher cellularity, greater levels of atypia, and lower detection of mitoses. The perception of increased cellularity can perhaps be explained by the increased depth of imaging (roughly 10 microns, compared to the 4–5 micron thickness of conventional slides), but also contributing to this impression could be the avoidance of fixation artifacts such that the interstitium appears more homogeneous or that the topographic texture unique to MUSE lends to a perception of depth not represented on H&E. The discrepancies seen for atypia and mitoses, on the other hand, could likely be due to the fact that these aspects are more difficult to appreciate at 10X magnification, the power used in the current MUSE prototype. While assessment of cellular features and tumor grading appear to be more modest with MUSE at its current stage, the diagnostic accuracy nonetheless appears promising and may be best indicated for applications where rapid tumor typing (rather than precise grading) is needed, such as in intraoperative clinical settings.

One limitation of this study was the relatively small sample size for the number of cases available for the diagnostic accuracy analysis (27) and DNA and RNA integrity study (2). For example, only having a single case of anaplastic oligodendroglioma may have contributed to the low accuracy rate seen for this grade in Table 2. This also extends to the sequencing analysis data, where having a limited sample size makes it difficult to ascertain whether aspects such as amount of material, differences in cellularity, or degree of necrosis could be confounding variables. We are hoping to expand to include a larger cohort of specimens as well as a larger panel of reviewers to achieve greater statistical power behind our data. Secondly, the limitation of the current MUSE prototype solely having the capability of 10X magnification is a shortcoming compared to a light microscope’s capability for 40X magnification. As mentioned previously, higher magnification could potentially augment the ability to discern nuclear and cytoplasmic features such as atypia and mitoses. 

In summary, the results of this study are promising in terms of observing relatively high diagnostic accuracy for neuropathological specimens and molecular preservation in a novel rapid and tissue-sparing method of microscopy. These implications are important in the context of modern clinical practice, where pathology turn-around time is sensitive and precision medicine is increasingly demanding the allocation of intact, unfixed tissue for molecular testing. Furthermore, the MUSE system employs a simple optics configuration that is easy to operate, utilizes readily accessible staining reagents, and allows for digital or tele-pathology without the need to generate physical glass slides. As such, it has the potential to be implemented widely in clinical settings as well as in low-resource settings with limited histology infrastructure. We hope to further expand this study in the near future to include comparisons of MUSE images with standard preparations versus intraoperative evaluations using touch preparations and smears, in hopes of exploring the potential for greater applicability to neuropathological practice. 

## 5. Patents

The MUSE patent number is US9964489B2.

## Figures and Tables

**Figure 1 brainsci-14-00108-f001:**
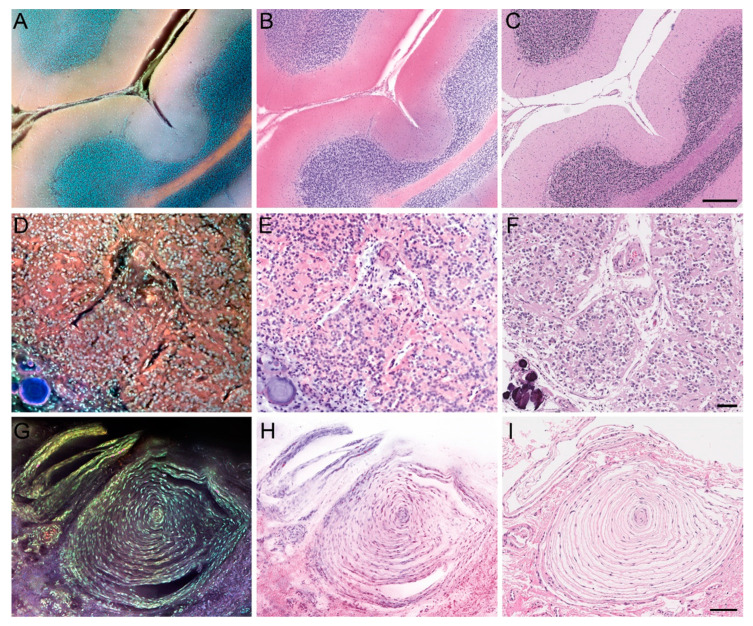
MUSE and H&E reconstruction of Normal Anatomical Structures. Left column: MUSE image. Middle columns: H&E reconstruction from the MUSE image. Right columns: Traditional H&E. (**A**–**C**) Cerebellum. (**D**–**F**) Pineal gland (note visible calcification). (**G**–**I**) Pacinian corpuscle. Scale bars: 300, 100, and 100μm.

**Figure 2 brainsci-14-00108-f002:**
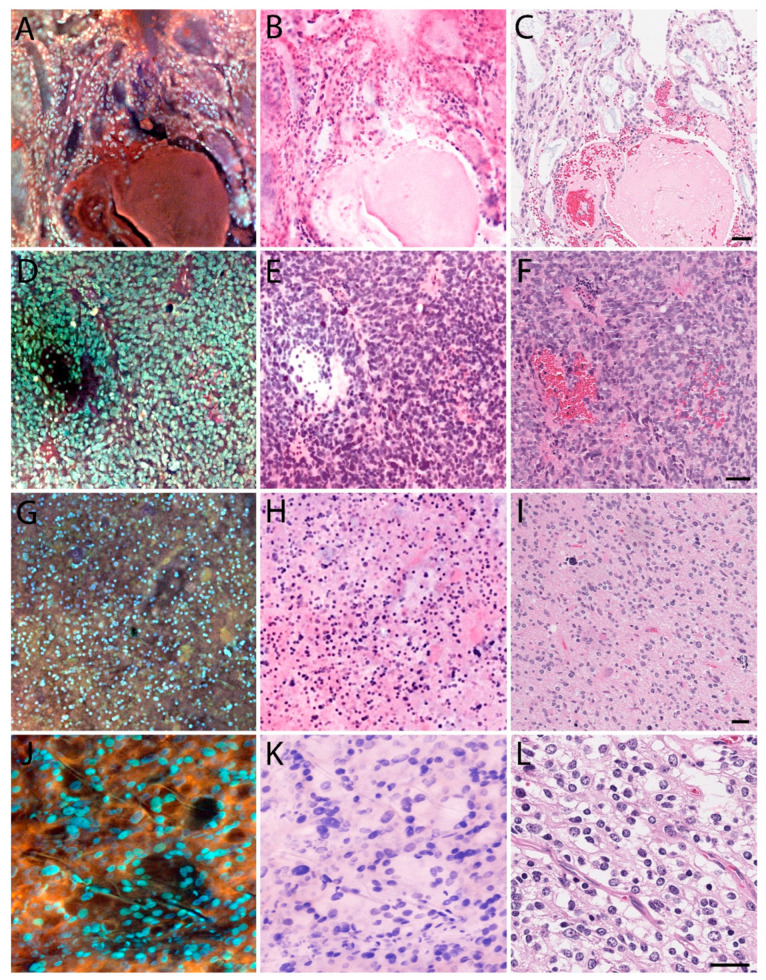
MUSE and H&E reconstruction on CNS tumor tissue. Left column: MUSE image. Middle columns: H&E reconstruction from the MUSE image. Right columns: Traditional H&E. (**A**–**C**) Myxopapillary ependymoma. (**D**–**F**) Anaplastic ependymoma. (**G**–**I**) Glioblastoma multiforme. (**J**–**L**) Oligodendroglioma. Images (**J**–**L**) are reprinted from Appendix A for Fereidouni et al., 2017 [4]. Scale bar: 50 μm.

**Figure 3 brainsci-14-00108-f003:**
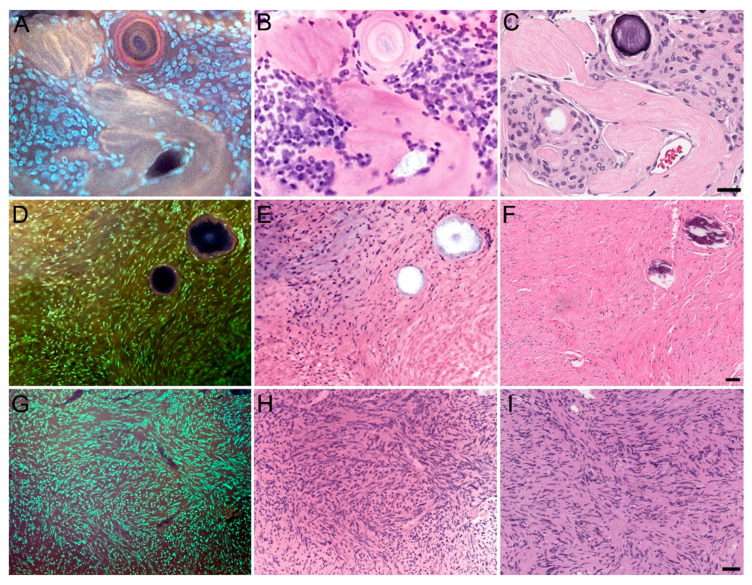
MUSE and H&E reconstruction on additional CNS tumor tissue. Left column: MUSE image. Middle columns: H&E reconstruction from the MUSE image. Right columns: Traditional H&E. (**A**–**C**) Meningioma. (**D**–**F**) Fibroblastic meningioma. (**G**–**I**) Schwannoma. Scale bar: 50 μm.

**Figure 4 brainsci-14-00108-f004:**
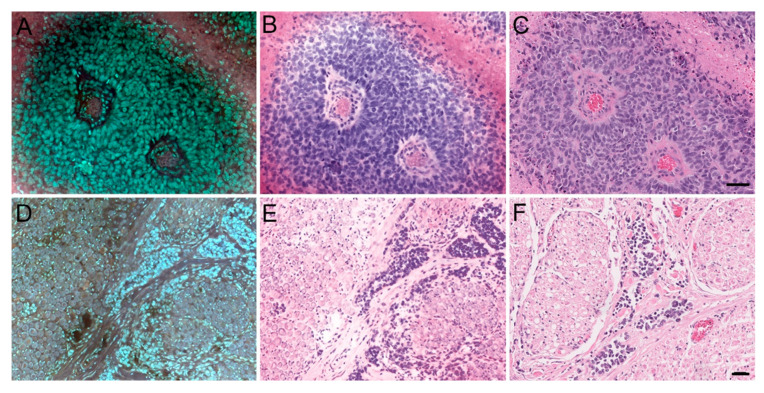
MUSE and H&E reconstruction on metastatic tumor tissue. Left column: MUSE image. Middle columns: H&E reconstruction from the MUSE image. Right columns: Traditional H&E. (**A**–**C**) Poorly differentiated metastatic carcinoma involving brain parenchyma. (**D**–**F**) Metastatic carcinoma wrapping around spinal nerve roots in a 50-year-old man (autopsy). Scale bar: 50 μm.

**Figure 5 brainsci-14-00108-f005:**
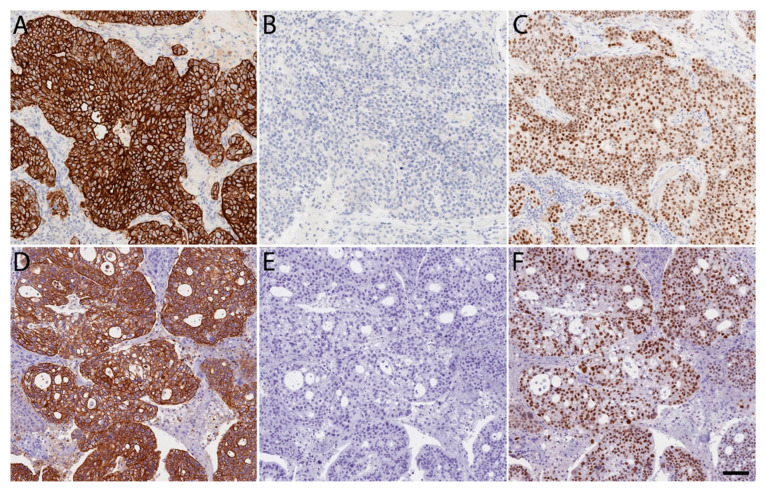
Immunohistochemical staining integrity is preserved following MUSE. Comparable quality of routine FFPE immunohistochemical staining of cerebellar metastatic Mullerian adenocarcinoma following standard 10% formalin fixation and without exposure to UV light (**A**–**C**) compared to identical immunohistochemical staining procedure after a short exposure to UV light during MUSE microscopy (**D**–**F**). (**A**,**D**) Cytokeratin 7; (**B**,**E**) Cytokeratin 20; (**C**,**F**) PAX-8. 200× magnification. Scale bar: 150 μm.

**Table 4 brainsci-14-00108-t004:** DNA and RNA yield and purity from MUSE-treated and fresh frozen tissues.

	DNA ng/μL (Total Yield, ng) [260/280 Ratio]		RNA ng/μL (Total Yield, ng) [260/280 Ratio]	
Sample	MUSE-Treated	Fresh Frozen	MUSE-Treated	Fresh Frozen
39N	28.7 (2009) [1.85]	16 (1120) [1.91]	49.2 (1968) [1.68]	113.5 (4540) [1.86]
53	113 (7910) [1.87]	85.3 (5971) [1.87]	195.4 (7816) [2.02]	250.5 (10,020) [1.97]
57T	46.5 (3255) [1.85]	30.8 (2156) [1.78]	75.5 (3020) [1.92]	101 (4040) [1.98]

Note: Nucleic acids concentrations were obtained using Qubit fluorometry. 260/280 nm absorbance ratios were obtained using Nanodrop spectrophotometry; values ≥ 1.80 were considered pure for DNA, and values ≥ 2.0 were considered pure for RNA.

**Table 5 brainsci-14-00108-t005:** DNA and RNA sequencing quality metrics obtained using ion semiconductor sequencing.

			Sample 39N		Sample 53	
	Metric	Pass Metric	MUSE-Treated	Fresh Frozen	MUSE-Treated	Fresh Frozen
**DNA**	Mapped reads	≥5,000,000	12,334,171	14,797,491	11,613,269	9,929,415
	Mean read length (bp)	≥75	115	114	112	112
	Uniformity (%)	≥90	95.9	96.4	97.5	97.7
	Mean target coverage	≥800	3703	4389	3426	2922
	On-target coverage (%)	≥85	96.0	95.9	96.1	96.2
	C > T transitions (%)	<8.0	5.7	5.8	5.8	5.8
**RNA**	Mapped reads	≥500,000	969,724	764,965	1,302,634	1,207,136
	Mean read length (bp)	≥60	108	93	111	109

Note: Shown in gray shading are the metric thresholds indicative of high-quality sequence data. These data were obtained following library preparation using an amplicon-based method and sequencing using a GeneStudio S5 sequencer (Thermo Fisher Scientific).

**Table 6 brainsci-14-00108-t006:** DNA sequencing quality metrics obtained using fluorescence-based sequencing.

	Sample 53	
Metric	MUSE-Treated	Fresh Frozen
Mapped reads	942,438,994	1,016,643,517
On-target reads	457,723,211	479,992,813
On-target duplicate reads	215,608,334	207,313,493
Mean target coverage	939X	989X

Note: hese data were obtained following library preparation using an enrichment-based method and sequencing using a NovaSeq 6000 sequencer (Illumina).

## Data Availability

The data presented in this study are available on request from the corresponding author. The data are not publicly available due to the occasional inclusion of PHI, which can be edited out on a per-case basis.

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
