# Peer review of "Neuropathological Applications of Microscopy with Ultraviolet Surface Excitation (MUSE): A Concordance Study of Human Primary and Metastatic Brain Tumors"

_brainsci, 2024, doi:10.3390/brainsci14010108_

Round 1

Reviewer 1 Report

Comments and Suggestions for Authors

Abstract: The statement "MUSE images were generated from surgical biopsy samples of primary and metastatic brain tumor biopsy samples (n=67)" implies that the study evaluated in 67 independent cases; however, the true number is noted in the M&M section and should be 30 which includes the 3 normal structures.  Please adjust the text to more accurately reflect the number of unique patients evaluated in this study.

Intro: 

lines 44-49: I recommend softening the language used to describe traditional pathology methods.  I dont agree that its excessively time intensive and time consuming compared to the neurosurgical procedure necessary to acquire the tissue.   It has been a relatively low-cost approach for decades.  I don't feel its necessary to diminish a proven technology that is globally adopted in order to introduce a novel technology such as MUSE.  They both have uses as the study itself highlights. 

Lines 58-64 describes the general MUSE approach with line 68 suggesting a pilot study; however, the pilot study relies on previously frozen samples but freezing samples before MUSE is not described in 58-64. Please clarify if freezing samples is a standard part of the processing for MUSE and if so, then the authors should describe that in lines 58-64. 

Methods: 

lines 92-94 introduce the time required to scan a single field but the authors do not provide any information on how long it takes to complete scanning a single case. Further in the manuscript a total scan time of 5 minutes is noted per cases and that should be included in the M&M section as well.

Results:

table 2: Given the clarity of the images provided in Figures 1-4, its unclear to this reviewer why GBMs, AO3, and M2s proved so diagnostically challenging.  The authors should provide representative images of discordant cases in the supplemental text along with some accompanying explanations for the challenges faced by the study pathologists. Also examples of MVP, necrosis, and mitoses as well as how they appear on standard H&Es vs MUSE H&Es are also required. 

Discussion:

The authors should provide more details with respect to the challenges faced by pathologists that led to the discordant diagnoses (GBM, AO3, MN2). For example, were necrosis and MVP not readily identifiable on H&Es constructed from MUSE. 

Reviewer 2 Report

Comments and Suggestions for Authors

The manuscript by Vivian Tang and collaborators, entitled "Neuropathological applications of microscopy with ultraviolet surface excitation (MUSE): a concordance study of human primary and metastatic brain tumors", presents an interesting work comparing Microscopy with ultraviolet surface excitation (MUSE) with traditional hematoxylin-eosin (H&E) staining in pathologic diagnosis.

Nowadays, brain tumor diagnosis are mainly conducted through H&E slides. This technique requires a long sample preparation procedure, but also a tissue-consuming method that can prevent further analysis of the same tissue. 

The need underlying this work is the necessity to develop a novel intraoperative protocol possibly with shorter and less invasive preparation of the sample.

The authors propose MUSE as a potential solution to this problem showing how the MUSE technique allows eliminating fixation & sectioning step while maintaining an intact tissue that can be further processed to obtain molecular biology data.

The work shows a high degree of diagnostic accuracy making MUSE a promising tool allowing preservation of the sample, compatibility with molecular biology technique after MUSE imaging, high-resolution images, and correct diagnosis avoiding fixation artifacts.

Another potential application of MUSE could be exploited in an intraoperative clinical setting where rapid tumor typing is required more than precise tumor grading.

Another implementation that could improve MUSE capabilities could be using higher magnification to implement the capabilities of MUSE in detecting Mitoses, atypia, and cellularity.

The document is clearly written and easy to read. The method description, results and discussion are easy to follow for a person not an expert in the field. One suggestion is to be careful to explain all the acronyms, even if these are popular (e.g. “H&E” line 34; “IHC” line 100).

Another comment relates to lines 200-212 the accuracy is not well explained. It is not clear whether it is 83-94% and so on. I suggest to write this part more in detail.
